# Scoping review protocol: is there a role for physical activity interventions in the treatment pathway of bladder cancer?

Sneha Mehrotra,[1] Megan Rowland,[1] Hanyu Zhang,[1] Beth Russell,[1] Louis Fox ![ORCID],[1] Katharina Beyer,[1] Elke Rammant,[2] Nicola Peat,[3] Mieke Van Hemelrijck ![ORCID],[1] Cecilia Bosco[1]

[1]Translational Oncology and Urology Research, King's College London, London, UK
[2]Human structure and repair, Ghent University, Ghent, Belgium
[3]Physiotherapy Dept - Guy's Cancer Centre, Guy's and St Thomas' NHS Foundation Trust, London, UK

**Correspondence to**
Dr Mieke Van Hemelrijck;
mieke.vanhemelrijck@kcl.ac.uk

## ABSTRACT

**Introduction** Patients with bladder cancer (BC) have been found to have worse experiences than those with other cancers which may partly be due to impact on quality of life. Currently, little is known about the impact of physical activity (PA) on BC outcomes. This scoping review aims to identify what interventions are available, their reported efficacy and feasibility, and a description of potential underlying biological mechanisms for their effects.

**Methods and analysis** Preferred Reporting Items for Systematic Reviews and Meta-Analyses Scoping Review (ScR) guidelines and the Levac methodology framework will be followed/used. Electronic databases will be searched (MEDLINE, EMBASE, the Cochrane Library, PsycInfo and Health, OpenGray). Two independent reviewers will screen all abstracts and titles and during a second stage and full-text publications for inclusion. All studies describing PA (as an existing lifestyle or as part of an intervention programme) during BC management will be included. Study characteristics will be recorded; qualitative data will be extracted and evaluated using the Donabedian framework. Quantitative data will be extracted and summarised. A further consultation step will be carried out with patients, their family members and healthcare professionals.

**Ethics and dissemination** Results will be disseminated through a peer-reviewed publication. Through the consultation step, we will ensure that findings will reach a wide audience and recommendations can be made for future development of PA interventions for patients with BC. Data used will be from publicly available secondary sources, and the consultation step will be carried out as part of patient and public involvement so this study does not require ethical review.

## INTRODUCTION

Bladder cancer (BC) is the ninth most common malignancy worldwide. There are approximately 10 300 new BC cases diagnosed in the UK every year which equates to about 28 new diagnoses every day.[1] BC accounts for 3% of all new cancer cases, and is the eighth most common male and 16th most common female cancer.[2]

The 2015, National Institute for for Health and Care Excellence (NICE) guidelines

### Strengths and limitations of this study

► This is a novel scoping review to understand what types of physical activity interventions exist within the management of bladder cancer.
► This review will extend the current reviews as it focuses on all patients with bladder cancer, both observational and randomised trial data, and assesses evidence for potential underlying biological mechanisms.
► Stakeholders including patients, their family members, urologists, oncologists, physiotherapists will be involved throughout the study.
► The identification and synthesis of data will also cover the grey literature.
► It is possible that our review will not include all articles which have been published in every journal as some may not be accessible.

for BC diagnosis and management stated,[3] 'There is thought to be considerable variation across the NHS in diagnosis and management of BC and provision of care to people who have it. There is evidence that the patient experience for people with BC is worse than that for people with other cancer.' The cause of the poor patient experience is multifactorial—but is mainly driven by the effects of cancer diagnosis and treatment on health-related quality of life (HRQoL).[4] About 30% of patients present with muscle-invasive BC (MIBC), although up to 45% of those presenting with non-muscle-invasive BC (NMBC) will subsequently progress to MIBC. Treatments for NMBC and MIBC differ, but both come with a variety of side effects, as well as invasive investigations and frequent follow-up due to the risk of recurrence and metastasis.[5]

Despite the burden of BC on different HRQoL domains,[4] relatively few HRQoL studies have been conducted in patients with BC compared with other tumour groups—so that it is also unclear how this may differ

between those with NMBC and MIBC.[6 7] A contributing factor for this lack of knowledge is the small investment in BC research: in the UK, the total annual research spend on BC is only £216 per new patient compared with £561 for prostate cancer.[8]

## STUDY RATIONALE

Physical activity (PA) interventions have been introduced for cancer patients as they are thought to contribute to better treatment outcomes and increase HRQoL.[9] More specifically, it has been suggested that integrating exercise training with standard cancer care and treatment may improve disease-related physiological and psychological outcomes in patients, as it helps to reduce drug toxicities and increases treatment completion rates.[10] For example, an analysis by Holmes et al in the Nurses' Health Study highlighted that engagement in more than 9 MET hours/week of PA following breast-cancer diagnosis was inversely associated with breast cancer-specific mortality risk.[11] A recent systematic review showed that the greater intensity PA interventions were associated with a greater beneficial effect on HRQoL which included the assessment of factors such as social functioning and fatigue.[12] Several mechanisms such as enhanced immune response, body composition, tumour vascularisation and tryptophan metabolism regulation have been suggested to explain the potential benefits of PA interventions in patients with cancer.[10]

In contrast to some other cancers, the literature on the potential positive effects of PA on BC management has not been addressed comprehensively across the treatment pathway, with currently one review focusing specifically on those patients who have undergone a radical cystectomy which is only affecting a subgroup of patients with BC.[13]

## STUDY OBJECTIVES

With this scoping review, we aim to investigate what information is currently available on the effects of PA on the various stages of the BC treatment pathway. This will help identify the gaps in the current work on the influence of PA on BC management. We will describe types and duration of PA (as an existing lifestyle or as part of an intervention programme), the reported effects on clinical outcomes and HRQoL, the feasibility of PA interventions and describe the proposed biological mechanisms (as this may also inform the target population and PA intervention design). Data from all types of available studies—mainly observational and randomised studies—will ultimately help us design a PA intervention which is beneficial for patients with BC, but also acceptable and feasible for all stakeholders.

## PROTOCOL DESIGN

Methods for this scoping review were developed based on the Joanna Briggs Institute guidelines[14] and more specifically the methodological guidelines developed by Levac et al,[15] which describe the below six framework stages. Preferred Reporting Items for Systematic Reviews and Meta-Analyses-ScR extension for scoping reviews[16] will be followed to ensure that all suggested items are reported.

### Stage 1: identifying the research question

Through consultation with the clinical research team, the overall research questions are defined as:
1. What type of PA (lifestyle measurements or PA interventions) has currently been reported to affect clinical outcomes and/or HRQoL for patients with BC?
2. What effects of PA (interventions) on clinical outcomes and HRQoL for BC have been reported?
3. What information is available on the feasibility of existing PA interventions?
4. Is there evidence to support underlying biological mechanisms for the effect of PA on BC development/outcomes?

### Stage 2: Identifying relevant studies—search strategy

The following electronic databases will be searched from inception until the date in which the searches will be performed (until November 2019): MEDLINE (using the PubMed interface) and Ovid Gateway (Embase and Ovid). The Cochrane Library and OpenGray will also be searched. The search strategies will be evaluated using the Peer Review of Electronic Search Strategies guidelines.[17]

Search terms have been determined through researcher input and researching the current available literature to help guide the selection of terms, ensuring they are broad enough to capture any PA intervention and BC study. Our scoping review will analyse both quantitative and qualitative data on PA and BC (see search strategy in online supplementary appendix). To ensure that all relevant information is retrieved, relevant grey literature sources will be searched.

### Stage 3: study selection

Studies will be considered for inclusion if they assess clinical/HRQoL outcomes in patients who have a primary diagnosis of BC and PA has been measured (either as part of their lifestyle or as part of an intervention in the treatment pathway). Studies will be excluded if the publication is not available in English. All papers derived from the digital search process will be uploaded to a reference management software (Endnote). From these references, we will then document the exclusion process of the studies; initially excluding irrelevant studies based on title alone, then based on abstracts. Two review authors will screen the studies independently, and any lack of consensus will be discussed with a third review author. After screening titles and abstracts, the full articles will be read and considered for the review also by two independent reviewers; those articles excluded will have recorded

evidence as to why this was necessary. For studies that have multiple publications of the same outcome(s) reported, the one with the longest follow-up will be selected. If older publications refer to articles, those included may be accessed to clarify methods if needed.

## Stage 4: charting the data

Two independent reviewers will conduct this process. The data extraction table produced will include at least the following headings:

1. Author.
2. Year of publication.
3. Country where the study was published/conducted.
4. Aims/purpose.
5. Study population and study size.
6. Study design (eg, observational, randomised controlled trial).
7. PA type +details (eg, lifestyle, intervention).
8. Duration of intervention.
9. Outcomes.
10. Key findings that relate to scoping review objectives.

## Stage 5: collating, summarising and reporting the results

For our scoping review, the studies identified will be analysed using both qualitative and quantitative methods. An overview of the research will be displayed through all the findings.

In terms of *qualitative aspects*, all the reported insights will be deductively coded into a conceptual model that is taken from the Donabedian conceptual framework to determine the advantages of PA for patients with BC.[18] This framework will be used to assess the quality of care and thus the effectiveness of the PA interventions reported through three domains: structure, process and outcome. By assessing interventions through three different domains, we will be able to identify the different strengths of PA interventions and use this to guide the design of future feasibility trials. *Structure* will regard the organisation, resources and equipment available and needed for the interventions. *Process* will evaluate how the intervention is implemented and the methods used by healthcare professionals when delivering the intervention and the effect this has on the patient's HRQoL. *Outcome* will address the overall mental state and feelings of patients taking part in the interventions.[18]

The *quantitative outcomes* will be assessed using information on the following measurements:

▶ Survival or recurrence rates/response to treatment/ HRQoL measures: to inform on efficacy of the PA lifestyle/intervention as well as patient selection.
▶ Adherence to the intervention: indication of practicality, acceptability and feasibility of PA interventions.

Given the nature of this scoping review, we will not be explicitly performing a risk of bias assessment as usually required for quantitative systematic reviews[16] because of the following four reasons.[1] We will not compare the clinical outcomes of any clinical studies, but only describe interventions used and their potential effects.[2]

Argumentative and qualitative data do not lend themselves to the risk of bias assessment common in quantitative systematic reviews.[3] We are likely to include many different types of studies which evaluations would have to be type-specific, and[4] quality assessments are likely be very difficult to standardise between reviewers and which could potentially add unnecessary bias into our own analyses."

## Stage 6: consultation—patient and public involvement

This scoping review is a first phase in a multistage research programme[19] aimed at developing a feasibility PA intervention for patients with BC. To ensure that our assessment of the existing evidence for the implementation of PA interventions in the BC treatment pathway identifies the right target population and format/timing of a PA intervention, we also aim to include a consultation phase in this scoping review. The results from this scoping review combined with the consultation phase will then lead to development of a PA intervention that it can be implemented in standard care.

This consultation phase is part of our Patient and Public Involvement strategy as we will work actively in partnership with patients, their family members and healthcare professionals to plan and design future PA interventions for patients with BC.[20] More specifically, we will run focus groups with patients with BC and their families, healthcare professionals, physiotherapists and behavioural scientist to help identify whether the results of the scoping review truly reflect the needs and expectations of all stakeholders involved. We will invite and recruit participants for this consultation phase through our ongoing collaborations with patient advocacy organisations and healthcare professional organisations (eg, Action Bladder Cancer UK, Fight Bladder Cancer, British Association of Urological Surgeons, British Uro-Oncology Group).[21]

## DISSEMINATION AND ETHICS

As outlined above, this scoping review with a consultation phase will constitute the first stage in a multistage research programme aimed at developing a feasibility PA intervention for patients with BC. Results will be disseminated through a peer-reviewed publication. Through the consultation step, we will ensure that findings will reach a wide audience and recommendations can be made for future development of PA interventions for patients with BC.

As the scoping review methodology is based on reviewing and collecting data from publicly available materials, this study does not require ethics approval. To facilitate knowledge translation activities, we will liaise with relevant stakeholders through patient advocate and healthcare professional organisations. This consultation step will be carried out as part of patient and public involvement, so this study does not require ethical review.

**Acknowledgements** The authors gratefully acknowledge the helpful comments from the bladder cancer patient support group.

**Contributors** Design of protocol: SM, MR, HZ, BR, LF, KB, MVH, CB. Draft of manuscript: SM, MR, HZ, CB. Final approval of manuscript: SM, MR, HZ, BR, LF, KB, ER, NP, MVH, CB.

**Funding** We thank CRUK for their support (C45074/A26553).

**Competing interests** None declared.

**Patient consent for publication** Not required.

**Provenance and peer review** Not commissioned; externally peer reviewed.

**ORCID iDs**
Louis Fox http://orcid.org/0000-0002-8280-1797
Mieke Van Hemelrijck http://orcid.org/0000-0002-7317-0858

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
