## [Reviewer comments · BMJ Open]

ARTICLE DETAILS

TITLE (PROVISIONAL)	Scoping Review Protocol: Is there a role for physical activity interventions in the treatment pathway of bladder cancer?
AUTHORS	Mehrotra, Sneha; Rowland, Megan; Zhang, Hanyu; Russell, Beth; Fox, Louis; Beyer, Katharina; Rammant, Elke; Peat, Nicola; Van Hemelrijck, Mieke; Bosco, Cecilia

VERSION 1 – REVIEW

REVIEWER	Beth Hunter, PhD, ORT/L University of Kentucky, USA
REVIEW RETURNED	19-Aug-2019

GENERAL COMMENTS	Great protocol. A couple of questions and comments: - how will grey literature be discovered and accessed? -page 7 line 47 "organ digital search" is this a typo?
---

REVIEWER	Brant Inman Duke University USA
REVIEW RETURNED	02-Sep-2019

GENERAL COMMENTS	This is a protocol for a scoping review describing the plan for a systemic review of the role of physical activity during bladder cancer management on various outcomes, including: physical activity intervention metrics (practicality and acceptability of, as well as adherence to the intervention), disease related metrics (health-related quality of life (HRQoL), response to treatment, recurrence rates, and survival). A "consultation phase" is also planned, and described at a high level as a corroboration of the findings from the above review through focus group discussions with multiple participants (patients, family members, healthcare professionals, physiotherapists, etc.) with the intent to devise a physical activity intervention for further study. This is the first of a multi-phase project with the broad goal of developing a physical activity intervention for patients with bladder cancer. Strengths -Relevant and concrete research questions surrounding the literature assessing physical activity in the bladder cancer space-Specific search queries planned for review of available literature from multiple databases-Plan to utilize PRISMA-ScR framework for standardized approach to literature search and evaluation-Quality-control plan to evaluate literature for relevance to the research questions
---

	Weaknesses  -The dates of the study are not included in the manuscript -The researchers describe their framework for evaluating the quality of care and effectiveness of the physical activity interventions described in the literature, but this framework does not explicitly indicate how the available data will be analyzed for weaknesses or bias (Stage 4), if at all. As indicated in the PRISMA-ScR checklist, this is an optional, and likely important step in this project. Patients treated for bladder cancer tend to be elderly and with varying levels of comorbidities and corresponding fitness. This likely would play a role in the applicability of any single study on the broader bladder cancer cohort. -The discussion around the importance of the topic weighs heavily on HRQoL concerns in bladder cancer. However, there is no discussion around which specific HRQoL metrics are of particular importance or the hypothesized relationship to physical activity. It is possible that physical activity proxied by age, education level, or some other variable that is correlated to HRQoL measures. This should be considered during the literature review to inform the planning of the physical activity intervention, and specifically how that physical activity intervention will be assessed for effect on the outcomes of interest. -In the event that limited data is available to answer the research questions in the existing literature, it is not clear how the researchers plan to utilize the consultation phase or how the researchers will devise a physical activity intervention as part of their broader goal. -Consider a threshold for “limited data” which would then trigger a sub-analysis of the specific question in other malignancies to leverage findings in a disease process with similar patient demographics and treatment considerations.
--	--

REVIEWER	Tiago S. Jesus, Ph.D Global Health & Tropical Medicine, New University of Lisbon, Portugal
REVIEW RETURNED	07-Oct-2019

GENERAL COMMENTS	Thank you for the opportunity to review this scoping review protocol. I must declare upfront I'm not an expert in the subject matter, rather in the methodology. Hence, my reviews are only focused on methods issues or their report. General comments Overall, the scoping review method (speaking about the full text) is adequate in most cases. I pinpoint where not below. The consultation with stakeholders is a good, optional practice, which is included here. Two independent reviewers are used throughout, from what one can infer – stated in some parts, but not matched on others (abstract vs full text). The number of databases searched for is not large, but altogether can meet the purpose, especially as combined with other data sources and the grey literature, as indeed planned and reported.
--

What strikes me the most – surely a matter for substantial improvement – is the lack of match between the abstract and the full text in too many occasions. For instance, if I used only the abstract to ground my review, I would even say that the stated goal could not be properly be addressed by a scoping review. If the actual goals are those in the main text, my appraisal is quite the opposite.

My section-by-section review is provided below:

Abstract

The aim in the abstract: *“Currently, little is known about the impact of physical activity (PA) on BC outcomes. This scoping review aims to identify what associations have been found with lifestyle PA or PA interventions (including their efficacy and feasibility), as well as the potential underlying biological mechanisms for their effects.”*

And the aim in the full text states: *“we aim to investigate what information is currently available on  the effects of PA on the various stages of the BC treatment pathway. This will help identify the gaps in the current work  on the influence of PA on BC management....”*

Are they of the same study: the second (in the full text, including the content omitted) seems pretty logic for what a scoping review can provide, the first (in the abstract) clearly not. If I would see only the one in the abstract, I’d say that a scoping review, no matter how well conducted, would not be the suitable design to respond to that aim but rather a fully-fledged systematic review, with quality appraisal included. What I think is wrong, after reading both (quite different versions, stating different types of goals) is what you write in the Abstract. There are other instances in which the contents of the abstract and the full text quiet don’t match – I state them throughout.

Methods and Analysis: I think you mean PRISMA-ScR, not PRISM-ScR. It is also important to note that the PRISMA-ScR is a reporting guideline, Levack a method of conduct. You state as if they were for the same methodological purpose. In the main text you also mention Joanna Briggs, another method of conduct – appropriate to refer it here too, with the methods of conduct and reporting separated from one another.

Once again, the databases in the abstract do not match those in the full text.

“Two independent reviewers will screen all abstracts and titles and during a second stage and full-text publications for inclusion”. See

	my comment later on, you don't state the second part in the main text. "All studies describing PA (as an existing lifestyle or as part of an intervention programme), during BC management will be included". My comment: this is false; later in the main text, you mention only observational studies and RCTs. "qualitative and quantitative data will be extracted and summarised.". By reading the full-text, one understands you will use a known framework for part of this. As it is, it says little to nothing. "Ethics and dissemination": At instances, the content here says more than the full text. Why the acronym nearly in the last line? "Strengths and limitations of this study" The first point should more accurately and precisely reflect what is the intended output. "The identification and synthesis of data will also cover the grey literature – which is not as easily searchable as the peer-reviewed published literature". The second part is implicit. Last point: It would be much more specific to state that you only cover a few databases. Indeed, scoping reviews typically cover 5-10. Methods (protocol design) Page 8, line 29-35 "Stage 3: Study selection. The studies that will be included will be determined through the use of the PRISMA-ScR extension for scoping reviews (17)." How so? If the PRISMA-ScR is a reporting guideline, not a set of selection criteria – you need to define your own, as you begin to define in the next sentences. Something is clearly wrong with the sentence transcribed above. Page 8, line 32-45 Can you explicitly use a common format such as PICO or so and do it so itemized? Dates covered are not mentioned. And what
--	---

	about articles with abstract in English and full papers in other languages – included or excluded? Not clear. Page 8 line 54-onwards. Two independent reviewers also apply to the review of the full texts? As written, one can only be sure that it refers to the screening of titles and abstracts. Page 10, line 2 onwards. Re: deductively coded into a conceptual model that is “adapted” from the Donabedian conceptual framework. In which ways was/is it adapted? By whom? And which specific purposes? You then write definitions/components of the Donabedian’s SPO framework which fully resembles (at least has the main categories) the original model. So, why “adapted”? It seems an application of the original model to this purpose and subject in particular – this, alone, would not confer the meaning of an “adapted” model. Page 11, line 40 onwards. The contents in the “dissemination and ethics” do not fully reflect the respective content in the abstract which, at instances, is more complete, although an “abstract” it is. The non-match of content between the full text and the abstract is, unfortunately, all too common in the health research publications – and here way too present, while it must be addressed/prevented carefully by all stakeholders. Appendix – search strategy in which venue?
--	---

VERSION 1 – AUTHOR RESPONSE

Reviewers' Comments to Author:

Reviewer: 1

Reviewer Name: Beth Hunter, PhD, ORT/L

Institution and Country: University of Kentucky, USA

Please state any competing interests or state ‘None declared’: None declared

Great protocol. A couple of questions and comments:

- how will grey literature be discovered and accessed?

RE: Thank you for your comment. Gray literature will be discovered and accessed by running searches in OpenGray search engine and by hand searching references from relevant publications. We have now clarified this in the methods section of the protocol manuscript.

-page 7 line 47 "organ digital search" is this a typo?

RE: Thank you. Yes it was a typo and has now been corrected.

Reviewer: 2

Reviewer Name: Brant Inman

Institution and Country: Duke University, USA

Please state any competing interests or state 'None declared': None declared

This is a protocol for a scoping review describing the plan for a systemic review of the role of physical activity during bladder cancer management on various outcomes, including: physical activity intervention metrics (practicality and acceptability of, as well as adherence to the intervention), disease related metrics (health-related quality of life (HRQoL), response to treatment, recurrence rates, and survival). A "consultation phase" is also planned, and described at a high level as a corroboration of the findings from the above review through focus group discussions with multiple participants (patients, family members, healthcare professionals, physiotherapists, etc.) with the intent to devise a physical activity intervention for further study. This is the first of a multi-phase project with the broad goal of developing a physical activity intervention for patients with bladder cancer.

Strengths

-Relevant and concrete research questions surrounding the literature assessing physical activity in the bladder cancer space

-Specific search queries planned for review of available literature from multiple databases

-Plan to utilize PRISMA-ScR framework for standardized approach to literature search and evaluation

-Quality-control plan to evaluate literature for relevance to the research questions

RE: Thank you for your comments.

Weaknesses

-The dates of the study are not included in the manuscript

RE: Thank you, we have now clarified that in the methods section: "The following electronic databases will be searched from inception until the date in which the searches will be performed (until November 2019)".

-The researchers describe their framework for evaluating the quality of care and effectiveness of the physical activity interventions described in the literature, but this framework does not explicitly indicate how the available data will be analyzed for weaknesses or bias (Stage 4), if at all. As indicated in the PRISMA-ScR checklist, this is an optional, and likely important step in this project. Patients treated for bladder cancer tend to be elderly and with varying levels of comorbidities and corresponding fitness. This likely would play a role in the applicability of any single study on the broader bladder cancer cohort.

RE: This is indeed an additional step and, bias assessment is not mandatory in scoping reviews. We have now clarified the reasons not to perform this appraisal in the methods section in the following paragraph : “We will not be explicitly performing a risk of bias assessment as usually required for quantitative systematic reviews because of the following four reasons: (1) We will not compare the clinical outcomes of any clinical studies, but only describe interventions used and their potential effects; (2) argumentative and qualitative data do not lend themselves to the risk of bias assessment common in quantitative systematic reviews; (3) we are likely to include many different types of studies which evaluations would have to be type-specific; and (4) quality assessments are likely be very difficult to standardize between reviewers and which could potentially add unnecessary bias into our own analyses.”

-The discussion around the importance of the topic weighs heavily on HRQoL concerns in bladder cancer. However, there is no discussion around which specific HRQoL metrics are of particular importance or the hypothesized relationship to physical activity. It is possible that physical activity proxied by age, education level, or some other variable that is correlated to HRQoL measures. This should be considered during the literature review to inform the planning of the physical activity intervention, and specifically how that physical activity intervention will be assessed for effect on the outcomes of interest.

RE: Thank you for your comment. As noted by the reviewer we expect to find several interventions, amongst different groups (age, sex, comorbidities) and therefore likely different ways to assess HRQoL metrics. Therefore, as the purpose of the review is to identify the gap in the literature, we will report all types of interventions and metrics and later on together with the results of the focus groups interviews. We will summarise the information and suggest which metrics/surveys cover and measure more accurately the needs/HRQoL of bladder cancer patients in relation to exercise (whilst taking into account all these potential confounders).

-In the event that limited data is available to answer the research questions in the existing literature, it is not clear how the researchers plan to utilize the consultation phase or how the researchers will devise a physical activity intervention as part of their broader goal.

RE: Thank you for your comment. The purpose of the scoping review is to identify what has been published around this topic in general. Therefore, the research questions are not narrow nor specific. The results of the review will show if this topic has been studied and if there are PA interventions for people with bladder cancer, how these have been implemented, if patients are satisfied and if the

goals of improving bladder cancer specific outcomes as well as general quality of life have been achieved. Therefore, if the literature does not provide much information, the consultation groups will help us understand what patients would like to see/experience in a PA interventions, and the professionals (physicians, physiotherapists, nurses, psychologists and PA trainers) will help us set PA goals that take into account the patients expectations while having a positive effect on specific and general outcomes.

-Consider a threshold for “limited data” which would then trigger a sub-analysis of the specific question in other malignancies to leverage findings in a disease process with similar patient demographics and treatment considerations.

RE: Thank you. This is a possibility, but having scanned the literature available, there are sufficient studies available to perform the scoping review as proposed. We have continued our work since submitting this proposal back in August and therefore believe that setting a threshold for limited data is not required.

Reviewer: 3

Reviewer Name: Tiago S. Jesus, Ph.D

Institution and Country: Global Health & Tropical Medicine, New University of Lisbon, Portugal

Please state any competing interests or state 'None declared': None declared.

I provide my comments in a separate, attached document.

Thank you for the opportunity to review this scoping review protocol.

I must declare upfront I'm not an expert in the subject matter, rather in the methodology. Hence, my reviews are only focused on methods issues or their report.

General comments

Overall, the scoping review method (speaking about the full text) is adequate in most cases. I pinpoint where not below. The consultation with stakeholders is a good, optional practice, which is included here. Two independent reviewers are used throughout, from what one can infer – stated in some parts, but not matched on others (abstract vs full text). The number of databases searched for is not large, but altogether can meet the purpose, especially as combined with other data sources and the grey literature, as indeed planned and reported.

RE: Thank you for your comments.

What strikes me the most – surely a matter for substantial improvement – is the lack of match between the abstract and the full text in too many occasions. For instance, if I used only the abstract to ground my review, I would even say that the stated goal could not be properly be addressed by a scoping review. If the actual goals are those in the main text, my appraisal is quite the opposite.

RE: Thank you. We have now updated the abstract to reflect the contents from the main text.

My section-by-section review is provided below:

Abstract

The aim in the abstract: “Currently, little is known about the impact of physical activity (PA) on BC outcomes. This scoping review aims to identify what associations have been found with lifestyle PA or PA interventions (including their efficacy and feasibility), as well as the potential underlying biological mechanisms for their effects.”

And the aim in the full text states: “we aim to investigate what information is currently available on  the effects of PA on the various stages of the BC treatment pathway. This will help identify the gaps in the current work  on the influence of PA on BC management....”

Are they of the same study: the second (in the full text, including the content omitted) seems pretty logic for what a scoping review can provide, the first (in the abstract) clearly not. If I would see only the one in the abstract, I'd say that a scoping review, no matter how well conducted, would not be the suitable design to respond to that aim but rather a fully-fledged systematic review, with quality appraisal included. What I think is wrong, after reading both (quite different versions, stating different types of goals) is what you write in the Abstract.

RE: Thank you for this very relevant observation. We have now corrected this discrepancy and both abstract and main text state the same aims (taking into account the abstract word count limitation)

There are other instances in which the contents of the abstract and the full text quiet don't match – I state them throughout.

Methods and Analysis

I think you mean PRISMA-ScR, not PRISM-ScR. It is also important to note that the PRISMA-ScR is a reporting guideline, Levack a method of conduct. You state as if they were for the same methodological purpose. In the main text you also mention Joanna Briggs, another method of conduct – appropriate to refer it here too, with the methods of conduct and reporting separated from one another.

RE: Thank you. Typos have been corrected, and guidelines referred to as guidelines and methods of conducts also as such. We will follow Levack’s method. We take into account some of JBI recommendations. PRISMA-ScR will be followed in order to make sure we report all necessary items as suggested in the reporting guideline.

Once again, the databases in the abstract do not match those in the full text.

RE: Thank you. We have now corrected this discrepancy and both abstract and main text state the same databases

“Two independent reviewers will screen all abstracts and titles and during a second stage and fulltext publications for inclusion”. See my comment later on, you don’t state the second part in the main text.

RE: Thank you. We have now corrected this in the main text.

“All studies describing PA (as an existing lifestyle or as part of an intervention programme), during BC management will be included”.

My comment: this is false; later in the main text, you mention only observational studies and RCTs. “qualitative and quantitative data will be extracted and summarised.”. By reading the full-text, one understands you will use a known framework for part of this. As it is, it says little to nothing.

RE: Thank you. We have now corrected this in the main text. We have also clarified in the abstract that a framework will be used to extract the qualitative data.

“Ethics and dissemination”: At instances, the content here says more than the full text. Why the acronym nearly in the last line?

RE: Thank you. This has been corrected and further clarified in the main text.

“Strengths and limitations of this study”

The first point should more accurately and precisely reflect what is the intended output.

RE: thank you. This has now been corrected to reflect the intended aim.

“The identification and synthesis of data will also cover the grey literature – which is not as easily searchable as the peer-reviewed published literature”. The second part is implicit.

RE: Thank you. This has now been deleted from the bullet point.

Last point: It would be much more specific to state that you only cover a few databases. Indeed, scoping reviews typically cover 5-10.

RE: Thank you. We have extended the search to at least 5 databases.

Methods (protocol design)

Page 8, line 29-35

“Stage 3: Study selection. The studies that will be included will be determined through the use of the PRISMA-ScR extension for scoping reviews (17).”

How so? If the PRISMA-ScR is a reporting guideline, not a set of selection criteria – you need to define your own, as you begin to define in the next sentences. Something is clearly wrong with the sentence transcribed above.

RE: Thank you. We have now corrected this.

Page 8, line 32-45

Can you explicitly use a common format such as PICO or so and do it so itemized? Dates covered are not mentioned. And what about articles with abstract in English and full papers in other languages – included or excluded? Not clear.

RE: Thank you. As stated by Levac, “Scoping study research questions are broad in nature as the focus is on summarizing breadth of evidence”. Therefore the PICO format does not apply to scoping reviews. Nonetheless, we have presented several questions around the topic in order to direct and clarify the focus of the review.

We have now added dates in the methods section and regarding English language the phrase “Studies will be excluded if the publication is not available in English” was already included in the manuscript under the “Stage 3: study selection” section.

Page 8 line 54-onwards.

Two independent reviewers also apply to the review of the full texts? As written, one can only be sure that it refers to the screening of titles and abstracts.

RE: Thank you this has now been clarified in the main text.

Page 10, line 2 onwards.

Re: deductively coded into a conceptual model that is “adapted” from the Donabedian conceptual framework. In which ways was/is it adapted? By whom? And which specific purposes?

You then write definitions/components of the Donabedian’s SPO framework which fully resembles (at least has the main categories) the original model. So, why “adapted”? It seems an application of the original model to this purpose and subject in particular – this, alone, would not confer the meaning of an “adapted” model.

RE: Apologies for the confusion, the word “adapted” has been changed to “taken”.

Page 11, line 40 onwards.

The contents in the “dissemination and ethics” do not fully reflect the respective content in the abstract which, at instances, is more complete, although an “abstract” it is. The non-match of content between the full text and the abstract is, unfortunately, all too common in the health research publications – and here way too present, while it must be addressed/prevented carefully by all stakeholders.

RE: Thank you. This has now been better described in the dissemination and ethics section to fully reflect the contents of the abstract.

Appendix – search strategy in which venue

RE: Thank you. This has now been clarified in the appendix.

VERSION 2 – REVIEW

REVIEWER	Tiago S. Jesus, Ph.D Global Health and Tropical Medicine, New University of Lisbon
REVIEW RETURNED	24-Oct-2019

GENERAL COMMENTS	Thank you again for the opportunity to review this paper, now in its revised form. The authors have addressed the issues mentioned. My only suggestion would be to simply add a little word in the Abstract - I insert that word here in CAPS: "This scoping review aims to identify what interventions are available, their REPORTED efficacy and feasibility (...)" This would turn the aim equivalent to that in the full text, and make the total difference for what a scoping review (without quality appraisal) versus an otherwise systematic review can provide.
---